# A Leak Zone Location Approach in Water Distribution Networks Combining Data-Driven and Model-Based Methods

**Marlon Jesús Ares-Milián [1], Marcos Quiñones-Grueiro [2], Cristina Verde [3] and Orestes Llanes-Santiago [1,\***

1   Automation and Computing Department, Universidad Tecnológica de La Habana José Antonio Echeverría, CUJAE, Marianao, La Habana 19390, Cuba; marlonj@icb.cujae.edu.cu
2   Institute for Software Integrated Systems, Vanderbilt University, Nashville, TN 37235, USA; marcos.quinones.grueiro@vanderbilt.edu
3   Instituto de Ingeniería, Universidad Nacional Autónoma de México, Mexico City 04510, Mexico; verde@unam.mx
*   Correspondence: orestes@tesla.cujae.edu.cu; Tel.: +53-72663280

**Abstract:** Model-based and data-driven methods are commonly used in leak location strategies in water distribution networks. This paper formulates a hybrid methodology in two stages that complements the advantages and disadvantages of data-driven and model-based strategies. In the first stage, a support vector machine multiclass classifier is used to reduce the search space for the leak location task. In the second stage, leak location task is formulated as an inverse problem, and solved using a variation of the differential evolution algorithm called topological differential evolution. The robustness of the method is tested considering measurement and varying demand uncertainty conditions ranging from 5 to 15% of node nominal demands. The performance of the hybrid method is compared to the support vector machine classifier and topological differential evolution approaches as standalone methods of leak location. The hybrid proposal shows higher performance in terms of location accuracy, zone size, and computational load.

**Keywords:** leak zone location; data-driven and model-based methods



## 1. Introduction

Water, a vital resource for humanity, plays multiple roles in everyday life: ranging from drinking water consumption to the fulfilment of daily tasks. Aiming to guarantee a ready supply of water for the continuously expanding urban areas, bigger and more convoluted water distribution networks (WDNs) have been implemented. As these distribution systems grow in size and complexity, the task of monitoring and diagnosing faults in their behaviour becomes more complex [1]. Leaks and pipe bursts are a common anomalous state in WDNs that generate significant waste worldwide in terms of water losses, wasted energy, and maintenance costs. As an average, 30% of the water pumped into urban areas is lost due to unattended leaks [2], in some cases reaching nearly 50% of the water pumped. Leaks can be classified into background leaks or pipe bursts [3]. Background leaks are commonly small in size and their effect is nearly imperceptible; therefore they represent a small percentage of water losses. Pipe bursts, on the other hand, are leaks of a bigger size and impact which can cause pressure drops in the network. These are the main cause of water losses in WDNs and its location is a current scientific problem that occupies the attention of several research groups in the world.

In order to detect the presence of a leak in a pipeline or set of pipes, hardware-oriented methods are employed. These are characterized by the use of specialized equipment based on infrared sensors, CCTV cameras, moisture sensors and, the most common, acoustic sensors, among others [2,4,5]. The analysis of the transient dynamics logged by most of these technologies allows an exact estimation of the leak location, and sometimes size [6,7]. The development in data logging technologies through wireless sensor networking and

the internet of things have rendered hardware-based leak location technologies more accesible [8]; however, the pipeline-level nature of these hardware-oriented methods reduces them to a local use. Employing these technologies to locate a leak by exploring a large area would be time consuming and result in high operational costs. Therefore, the exploration of every potential leak pipe in a WDN would be prohibitive [9].

Higher scale software-based leak location approaches have therefore been developed by several authors using the operational variables (i.e., pressures and flows) in the network [10–18]. These approaches have received significant attention due to the increasing development and accessibility of SCADA systems, as well as other data acquisition technologies. Software-based leak diagnosis methodologies can be classified into two distinct groups: model-based strategies [15,16], and data-driven approaches [10,11,14]. Hybrid solutions have also been proposed by combining data-driven and model-based approaches [17].

Model-based strategies are centered around a mathematical model of the WDN that describes the relationship between its operational variables, taking into account the network's structure (topological layout, pipe dimensions, etc.). Leak diagnosis is then effected by comparing the model outputs with the measured variables in the network. Quiñones-Grueiro et al. [15] and Steffelbauer et al. [19] formulate the leak location task as an inverse problem, and identify the location of the leak by finding the optimum network parameters that maximize the similarities between the model output and the measured leak sample. The resulting optimization problem is then solved using the differential evolution (DE) algorithm in [19]; and a modified version of DE which considers the topological characteristics of the network is used in [15]. Li et al. [18] propose a more direct approach, using the network model to generate a set of sensitivity matrixes that characterize several leak scenarios in the network. The similarity between the leak samples and the simulated samples in the sensitivity matrixes is then evaluated in order to identify the simulation conditions (leak location and size) of the most similar simulated scenarios.

Model-based leak diagnosis strategies do not require historical data of all the network modes, i.e., leaks of different sizes and locations. However, a model of the WDN is not always available, and its development may be too expensive or complex. Furthermore, the quality of these approaches depends directly on proper model calibration [16]; which should include the modeling of uncertainties like variations in consumer demands, variations in pipe roughness and diameters due to old age, and sensor accuracy and noise.

Data-driven methodologies take advantage from the historical data from the network and develop data analysis strategies. These may include statistical analysis such as process control charts [10]; however, an increase in the use of machine-learning-oriented methodologies has taken place in the past years [11,14,20,21]. Romero et al. [14] and Zhou et al. [12] define the leak location as a classification problem, and solve it using deep learning techniques. Chen et al. [13] and Shekofteh et al. [22] implement random forests (RF) and artificial neural networks (ANNs), respectively, to solve the leak location problem as a hierarchical classification problem. Sun et al. [21] estimate the pressure values in every node of the network by applying Kriging spatial interpolation [23]. These estimated pressure values are then used to locate the leak applying linear discriminant analysis (LDA) and ANNs as classifiers. Other machine learning techniques have been used such as support vector machines (SVMs) [11,20] and Gaussian process regression [24].

On the contrary of the model-based approaches, data-driven methodologies only require knowledge of the structural properties of the network and historical data from a representative set of network operational modes. However, this advantage doubles as an obstacle, since historical data of all the network modes is rarely available. Therefore, synthetic data generated by a network model is often used for training these data-driven methodologies [11–13].

Moser et al. [17] present a hybrid strategy named error-domain model falsification, which diagnoses the leak sample by comparing it to a set of model predictions. This comparison is effected against a threshold that characterizes the variations (uncertainties) in

the network operational variables. This decision thresholds are generated from a statistical analysis of historical data. Chen et al. [13] and Zhang et al. [20] use clustering algorithms based on sensitivity matrixes generated by network models to group the network nodes into zones before locating the leaks.

A leak can appear in any location in the network. However, it is an accepted assumption in software-based methodologies to estimate the leak location strictly as a node [11,12,25]. This simplification of the leak location problem has nevertheless been insufficient due to the inherent uncertainties in the WDN operational variables; and the fact that, in most networks, the number of sensors is significantly smaller than the number of nodes. Therefore, finding the exact leak node with high accuracy is a difficult task. Several works have estimated a zone (a group of network nodes) instead of a single node as a leak location [11–13,15,20]. This increase in potential location size further simplifies the leak diagnosis, allowing higher performance. Once the location of the leak has been estimated to a relatively small area, hardware-oriented technologies can be used in order to identify the exact location of the leak at pipeline level.

Clustering is often used for generating the zones in the network. Zhang et al. [20] and Chen et al. [13], for example, use k-means clustering to group the network nodes with similar leak patterns by analyzing a sensitivity matrix. Quinones-Grueiro et al. [11], however, use k-medoids clustering to group the nodes according to their topological characteristics (shortest pipe distances between network nodes). Another zone generation approach consists in selecting multiple ranking candidates from a final solution. Such is the case of Zhou et al. [12] who, through a neural-network-based methodology, produces a location probability for every node in the WDN. The top 5 nodes with the highest leak location probabilities are then selected as the potential leak zone location. The topological characteristics of the network are also considered for leak zone generation by including the area near the estimated leak node. Quiñones-Grueiro et al. [15] construct a zone by also considering the nodes neighboring the identified leak node as potential leak locations. A neighboring node is defined as connected to the initial estimated location node through a single pipe (regardless of the pipe length). Li et al. [18] analyse the relationship between zone size and location accuracy by generating a zone with the nodes that fall within a threshold pipe distance (defined as the total pipe length in the shortest path between two nodes) from the initial estimated location. As expected, a higher pipe distance threshold reduces the estimated location size while increasing the accuracy of the location strategy.

The main goal of this work, and its main contribution, is the proposition of a hybrid methodology for the location of leaks in WDNs. This methodology consists of two stages: an initial data-driven stage in which a subzone of the WDN is identified as a potential leak location by means of a multiclass SVM classifier; and a second stage that improves the leak location estimated in the first stage through a model-based approach. This second stage is formulated as an inverse problem, and it is solved using a variation of the Differential Evolution (DE) algorithm called Topological Differential Evolution (TDE). The first stage is meant to work as a search space reduction for the inverse problem. This search space reduction is expected to improve leak location performance and reduce computational cost.

The proposed methodology is tested under different demand uncertainty conditions using the hydraulic model of the WDN in the city of Modena, Italy. The performance of the hybrid method is compared to the SVM classifier and TDE approaches independently as standalone methodologies.

This paper is structured as follows: Section 2 presents the theoretical bases for this study, as well as the methodology proposed for the location of leaks. Section 3 presents the Modena WDN as case study to evaluate the proposed strategy; furthermore, the characteristics of the developed simulations are detailed, and experimental design is explained. In Section 4, test results are presented and discussed, and the proposed method is compared to other leak location approaches. Finally, conclusions and recommendations for future works are presented.

## 2. Materials and Methods

### 2.1. Network Modeling

The hydraulic model of a WDN requires previous knowledge of its topological characteristics, as well as a detailed account of the physical principles that rule its behaviour. An approximated mathematical model can be developed based on the mass and energy balance equations of the network, assuming that flows are stationary, incompressible and permanent; constant fluid density is also assumed. This model is based around two main elements: pipes or links, which represent actual pipelines; and network nodes, or junctions, which represent important consumer takes, output and entry points in the network, and connections between pipes.

The steady-state mathematical model of a WDN is described by a set of equations based on the associated physical laws. The flow dynamics in the *j*th node of the network are modeled as follows [26]:

$$\sum_{i=1}^{n_j} q_{ij}^{in} - \sum_{i=1}^{n_j} q_{ij}^{out} = d_j; \quad for\ j = 1, 2 \dots N_T;  \tag{1}$$

where $N_T$ is the number of nodes in the WDN, $n_j$ is the number of pipes connected to node $j$, $q_{ij}^{in}$ and $q_{ij}^{out}$ represent input and output flow, respectively, at node $j$ through pipe $i$, and $d_j$ represents the consumer's demand at the node.

The energy balance equations of the model are formalized as follows:

$$\Delta H = \sum_{e=1}^{E} h_e + \sum_{f=1}^{F} h_f,  \tag{2}$$

which states that the head difference $\Delta H$ between any two nodes in the network is always the same for every path between the two nodes [26]. With $h_e$ being the head drops associated with $E$ elements in the path and $h_f$ representing head lifts in $F$ elements along the path. The head-flow relationship model in a pipe has been defined in different ways. Darcy-Weisbach's and Hazen-Williams's, presented below, are the most common:

$$h_d = Rq^\varrho,  \tag{3}$$

where $q$ is the flow in the pipe and $h_d$ is the head drop. $R$ is a coefficient that depends on the model used, which summarizes several network characteristics like pipe dimensions and roughness coefficients; and $\varrho$ usually has a value close to 2 [26].

Leaks are modeled as pressure-driven aggregated demands in the nearest node; the node flow dynamic presented in (1) is then redefined for the leak node $l$ as:

$$\sum_{i=1}^{n_l} q_{il} = d_l + f_l; \quad f_l = E_c h_l^{\xi};  \tag{4}$$

where $\xi = 0.5$ [27], $h_l$ is the pressure head at node $l$, and $E_c$ is an emitter coefficient, which summarizes the leak dimensions.

### 2.2. Leak Location Methodology

The methodology proposed for leak location is presented in Figure 1. A previous leak detection step is assumed following any of the procedures found in the literature [9,15,17,28]; therefore, only leaks detectable by the sensors installed in the network will be located. A set of measured hydraulic variables $\mathbf{m} \in \Re^{n_s}$ represents each of the leak samples, with $n_s$ being the number of sensors (**S**) installed in the network. The estimated leak location is generated in two consecutive stages: On the first stage, the WDN is partitioned into a set of candidate leak zones using agglomerative clustering [29], and taking into account the topological relationships between network nodes, presented in $\Lambda \in \Re^{N_T \times N_T}$. A

previously trained SVM multiclass classifier, combined with Bayes temporal reasoning [30], is then used to estimate a potential leak zone $\mathbf{Z}_C$. In this stage, a potential leak size range $\mathbf{E}_{\mathbf{C}range} = [E_{cmin}, E_{cmax}]$ is also estimated for each sample; and an analysis of dominant sensors ($\mathbf{S}_d$) is effected based on the estimated leak zone $\mathbf{Z}_C$. On the second stage, an estimated leak node is generated as the solution of an inverse problem using *Topological Differential Evolution* (TDE) [15], with $\mathbf{Z}_C$ and $\mathbf{E}_{\mathbf{C}range}$ as search space restrictions. Finally, temporal reasoning is applied to the leak location estimated as the solution of the inverse problem and extended to the nearby neighbor nodes following the procedure presented in Section 2.4.1, generating the estimated leak zone location $\mathbf{Z}_{ngh}$ for sample $\mathbf{m}$.

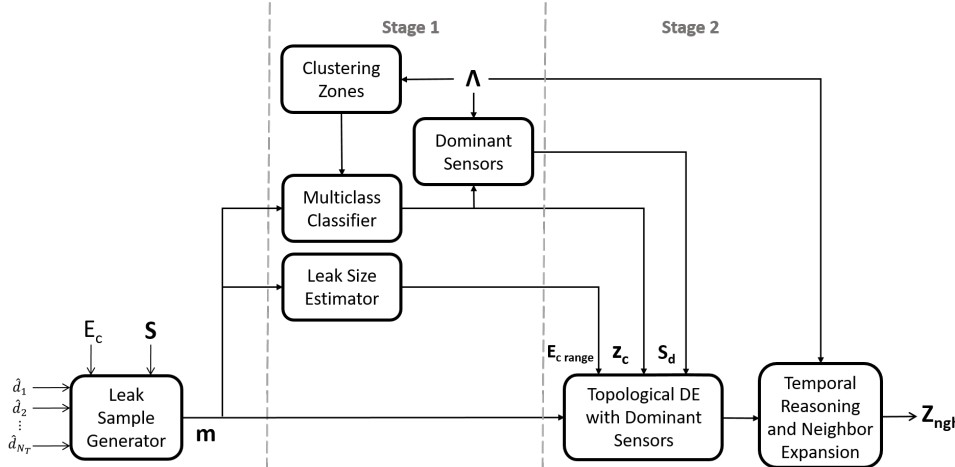

**Figure 1.** Leak Location Methodology.

The requirements for the implementation of the proposed methodology in a practical WDN application are presented below:

- A set of $n_s$ sensors installed in the network to capture the hydraulic information.
- A SCADA system for data acquisition and processing.
- A calibrated network hydraulic model.
- A processing unit (e.g., a personal computer (PC) or industrial PC) to implement and run the methodology.

The previous requirements can be fulfilled for many urban networks today, being the hydraulic model the main constraint due to the complexity of its calibration. The proposed methodology does not require a specific number, or type (pressure or flow), of sensors to be installed in the network, however, a large number of sensors will most likely result in a better performance. In spite of being a hybrid leak location methodology, historical data of the network behaviour is not necessary, since synthetic data generated by the network model can be used to train the multiclass classifier. Furthermore, the proposed methodology is completely unintrusive, not requiring any water supply reductions or interruptions for leak diagnosis.

### 2.3. Stage 1: Search Space Reduction and Dominant Sensor Selection

#### 2.3.1. Clustering Zones

On the first stage, a clustering algorithm is used to divide the network into a set of zones. In this work, agglomerative clustering [29] was selected due to its simplicity and effectiveness. Agglomerative clustering is a hierarchical clustering algorithm that generates a cluster dendrogram which goes from $N_C$ clusters at the bottom level to a single cluster at the top. This dendrogram is built by iteratively grouping a set $\mathbf{C} = \{c_1, c_2, \ldots, c_{n_c}\}$ of $n_c$ elements [29]. For WDN node grouping, every node in the network is considered as an element, and $n_c = N_T$, with $\mathbf{C} = \{1, 2, \ldots, N_T\}$. A distance (or similarity) metric $d(\cdot, \cdot)$ must be defined between network nodes in order to build the cluster dendrogram. In this case, the total pipe distance in the shortest path between two nodes is considered

as the similarity metric: $d(c_i, c_j) = \Lambda[i, j]$. Once the cluster dendrogram has been built, $n_z$ clustered zones can be produced, with $1 \leq n_z \leq N_C$, by selecting the corresponding level in the dendrogram [31]. By matching every leak sample with the clustered zone that contains the actual leak node, $n_z$ class labels are defined for the multiclass classifier.

### 2.3.2. Support Vector Machine Classifier

A Support Vector Machine (SVM) is selected as the classifier for the first step in the leak zone location methodology presented in this work. SVM classifiers are based on finding a hyperplane that optimally separates the two classes in training data set $D$ by maximizing the margin between the hyperplane and the support vectors [32]. The support vectors are defined as the datapoints from any of the classes that are closest to the separating hyperplane. Aiming to maximize the separating power of the hyperplane, data samples are projected into a higher dimensional space by means of the Kernel trick. Therefore, a kernel function $K(\cdot, \cdot)$ is defined between any two datapoints. The separating hyperplane between two classes is formalized as follows: $g(\mathbf{x}) = w^T \mathbf{x} + b$; where $w \in \mathbb{H}$ is a margin vector derived from the support vectors, with $\mathbb{H}$ being the dot product space of feature samples; and $b$ is an offset value. The position of the hyperplane is described by both $w$ and $b$, and $\mathbf{x}$ is a feature vector or data sample. The training of an SVM classifier is then reduced to finding the optimal values for $w$ and $b$ solving the optimization problem presented in (5) [33]:

$$\max_{w,b} [\sum_{i=1}^{n_t} a_i - \frac{1}{2} \sum_{i,j=0}^{n_t} a_i a_j g_i g_j K(\mathbf{x}_i, \mathbf{x}_j)];$$

$$\text{with } \sum_i g_i a_i = 0, \ 0 \leq a_i \leq C; \tag{5}$$

where $\mathbf{a} \in \Re^{n_t}$ are Lagrange multipliers; $D \in \{\mathbf{x}_i, y_i\}^{n_t}$ is the training dataset; and $C \in \Re$, $C > 0$, represents an error penalty that acts as an upper bound to limit the influence of the individual samples. For this work, the Radial Basis Function (RBF) Kernel, defined in (6), will be selected due to its non-linear nature, its small number of parameters, and its success rate in previous works [34].

$$K(\mathbf{x}_i, \mathbf{x}_j) = e^{-\gamma(\|\mathbf{x}_i - \mathbf{x}_j\|)^2}; \ \gamma > 0. \tag{6}$$

In order to ensure the maximum performance of the SVM classifier, a proper hyperparameter selection must be effected for $\{C, \gamma\}$ [20,34].

### 2.3.3. Bayes Temporal Reasoning

Analyzing leak samples over a time span $T_H \in \mathbb{Z}^+$, rather than at a single time instance, has shown to improve the leak localization accuracy [15]. Therefore, Bayes temporal reasoning, as defined in [21,30], is applied to the classification probabilities from a group of $T_H$ leak samples before estimating a leak location $\mathbf{Z}_C$.

### 2.3.4. Leak Size Estimation

For this work, the leak size range is assumed to have been previously estimated for every sample following the procedure presented in [15]. The potential leak range for sample $i$ is therefore estimated as $\mathbf{E}_{C_{range_i}} = [E_{ci} - 0.1, E_{ci} + 0.1]$, where $E_{ci}$ is the actual leak size.

### 2.3.5. Dominant Sensor Selection

In medium to large WDNs, a leak at a given node has often little to no effect in the hydraulic variables (flows and pressures) of the nodes located far away. Therefore, variations due to uncertainties in measurements at locations far away from the leak node can result detrimental to the leak diagnosis accuracy. In order to avoid this effect, Li et al. [18] define a set $\mathbf{S}_d \in [1, N_T]^{n_{sd}}; \mathbf{S}_d \subset \mathbf{S}$ of *dominant sensors* as the $n_{sd}$ sensors that best reflect the variations caused by a specific leak in the network, with $\mathbf{S} \in [1, N_T]^{n_s}$. For this work, a

set of dominant sensors was selected for every leak sample by considering the $n_{sd}$ sensors closest to the estimated leak zone $\mathbf{Z}_C$, following the procedure presented in Appendix A.

### 2.4. Stage 2: Leak Location as an Inverse Problem

An inverse problem consists in mapping the effect of the variation in model parameters on the input-output relationship of a given model [35]. Therefore, the solution of an inverse problem comes down to finding a parameter vector $\boldsymbol{\theta}$, such that the model output $\hat{\mathbf{y}}$ matches the real output $\mathbf{y}$ for a given input $\mathbf{x}$. The leak location task can be posed as an inverse problem by defining the model parameters $\boldsymbol{\theta} = \{E_c, \omega\}$ for a hydraulic model of the WDN, where the input vector $\hat{\mathbf{d}}_N \in \Re^{N_T}$ represents the nominal demands, and $\hat{\mathbf{m}}_{\mathbf{d}}$ is a vector of the model outputs measured by the dominant sensor. The leak location problem is then solved by finding the leak node $\omega$ in which a simulated leak of size $E_c$ maximizes the similarity between the output vector $\hat{\mathbf{m}}_d$ and the real leak sample $\mathbf{m}_d$ measured by the dominant sensors. Leak node estimation is then formalized as the optimization problem in (7):

$$
\begin{aligned}
\min_{\boldsymbol{\theta}} f_n(\mathbf{m}_d, M(\boldsymbol{\theta})), \quad &\boldsymbol{\theta} = \{E_c, \omega\}; \quad s.t. \\
&E_{cmin} \leq E_c \leq E_{cmax}, \\
&\omega \in \mathbf{Z}_C;
\end{aligned}
\tag{7}
$$

where $M(\boldsymbol{\theta}) = \hat{\mathbf{m}}_d$ represents the network numerical model. The parameter search space consists of: $[E_{cmin}; E_{cmax}]$, which defines the limits for the possible leak sizes; and $\mathbf{Z}_C$, which defines the topological search space (i.e., the potential leak location nodes). If the topological search space hasn't been restricted, every node in the network is a potential leak location, and $\mathbf{Z}_C = \{1, 2, \ldots, N_T\}$. The cost function $f_n$ is defined as the Euclidean distance between the real leak sample $\mathbf{m}_d$ and the simulated leak sample $\hat{\mathbf{m}}_d$ as presented in (8):

$$
f_n(\mathbf{m}_d, M(\boldsymbol{\theta})) = \sqrt{\sum_{k=1}^{n_{sd}} (|m_{d_k} - \hat{m}_{d_k}|)^2}.
\tag{8}
$$

The optimization problem presented in (7) is solved using a variation of the differential evolution algorithm called topological differential evolution (TDE). For its implementation, a mutation ($F$) and a crossover ($C_r$) coefficient must be properly selected. Furthermore, the following stop criteria must also be selected for the TDE algorithm: maximum iterations $g_{max}$, stagnation iteration limit $S_L$, and cost tolerance $\varphi$. A detailed description of the TDE algorithm is presented in Appendix B.

#### 2.4.1. Temporal Reasoning and Neighbor Expansion

Temporal reasoning is implemented for TDE by generating a leak location zone combining the estimated nodes over a time span $T_H$. Defining $\omega = \{\omega_1, \omega_2, \ldots \omega_{T_H}\}$, where $\omega_i \in \mathbb{Z}^+$ is the estimated leak node for sample $i$, an estimated leak zone by means of temporal reasoning is then defined as $\mathbf{Z}_T = \cup_{i=1}^{T_H} \omega_i$.

As stated earlier, leak zone generation can be implemented extending the estimated leak location to the nearby neighbor nodes [15,18]. Neighbor zone extension from the temporal-reasoning-derived estimated zone $\mathbf{Z}_T$ was implemented in this work. Assuming $\mathbf{Z}_T$ consists of $N_n$ nodes with $1 \leq N_n \leq T_H$, matrix $Z_m \in \{0, 1\}^{N_n \times N_T}$ is defined:

$$
Z_m[i, j] = \begin{cases} 1 & if \ j = \omega_i \\ 0 & otherwise \end{cases}.
\tag{9}
$$

Submatrix $\Lambda_Z$ contains the pipe distances from every node in the network to each node in $\mathbf{Z}_T$, and is calculated as follows:

$$
\Lambda_Z = Z_m * \Lambda
\tag{10}
$$

Finally, the estimated leak zone $\mathbf{Z}_{ngh}$ is generated by grouping the network nodes with a total pipe distance smaller than a threshold $\lambda$ from at least one of the nodes in $\mathbf{Z}_T$:

$$\mathbf{Z}_{ngh} = \{n \in \mathbb{Z}^+, n \in (1, N_T) | \Lambda_{Zi}(n) \neq \varnothing\}, \tag{11}$$

where

$$\Lambda_{Zi}(n) = \{i \in \mathbb{Z}^+, i \in (1, N_n) | \Lambda_Z[i, n] < \lambda\}. \tag{12}$$

*2.5. Performance Measures*

2.5.1. Leak Location Accuracy

Leak location accuracy is defined as the percentage of leak samples that were correctly located in the dataset:

$$ACC = 100 * \frac{\sum_{i=1}^{n_t} L_i}{n_t}, \tag{13}$$

with

$$L_i = \begin{cases} 1 & \text{if } \Omega_i \in \mathbf{Z}_{ngh_i} \\ 0 & \text{otherwise} \end{cases}; \tag{14}$$

where $\Omega_i$ is the actual leak node for sample $i$ and $\mathbf{Z}_{ngh_i}$ is the estimated leak zone for sample $i$.

2.5.2. Leak Zone Size

Two leak zone size indexes are defined. The first one is the number of network nodes in the estimated leak location:

$$ZNS_i = |\mathbf{Z}_{ngh_i}| \tag{15}$$

The second zone size index defined is the total pipe length in the zone and is formalized as follows:

$$ZPS_i = \frac{\sum_{j=1}^{N_T} \sum_{k=1}^{N_T} \beta_{jk} \Lambda[j, k]}{2}, \tag{16}$$

where

$$\beta_{jk} = \begin{cases} 1 & \text{if } j, k \in \mathbf{Z}_{ngh_i}, \text{ and } j \text{ and } k \text{ are neighbor nodes} \\ 0 & \text{otherwise} \end{cases}. \tag{17}$$

Note that $ZPS_i$ is calculated by dividing by 2 to account for the fact that $\Lambda$ is a symmetric matrix. A small leak size index, both in number of nodes and total pipe distance, is a desirable result. Smaller estimated leak zones reduce the time it takes to find the exact leak location in a subsequent step. Mean and standard deviation values are calculated for every dataset: $\{\overline{ZNS}, \sigma_{ZNS}\}$, $\{\overline{ZPS}, \sigma_{ZPS}\}$.

2.5.3. Computational Cost

Since online test time for the classifier is negligible in comparison to the cost of solving the optimization problem presented in (7), computational cost $CC$ is measured as the number of generations $g_{last}$ before reaching any of the stop criteria:

$$CC_i = g_{last_i} \tag{18}$$

Mean and standard deviation values are calculated for every dataset: $\{\overline{CC}, \sigma_{CC}\}$.

## 3. Case Study

*3.1. Modena Network*

The proposed leak location methodology can be implemented in any WDN with a properly calibrated hydraulic model. However, its full potential is best taken advantage of in medium-to-large scale WDNs, since the reduction of the search space considered in stage 1 of the methodology is practically unnecessary in small networks. Therefore, the proposed methodology was tested on the hydraulic model of the WDN in the city of

Modena, Italy (Figure 2), which presents characteristics common to most urban WDNs. It is a medium sized WDN with 317 pipes and 268 nodes [36]. It is gravity-fed by 4 reservoirs, therefore, no pumps are installed. This is a meshed WDN, with a relatively high number of connections per node, which results in higher correlation among the hydraulic variables at different nodes.

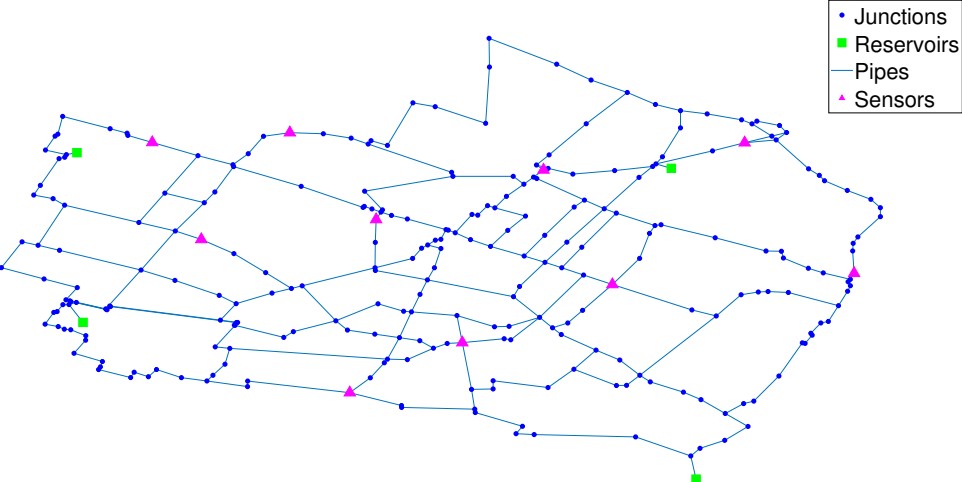

**Figure 2.** Modena WDN.

Full supervision of pressures and flows in every node and pipe of a WDN is rarely found in practice since it would result in excessive instrumentation costs. Therefore, a great number of leak diagnosis methodologies have been implemented by supervising only a limited number of variables [13,15]. For the purpose of this study, ten pressure sensors are installed in the nodes marked in Figure 2. Pressure sensors are often preferred over flow sensors due to their lower installation and maintenance costs. The optimal location for the pressure sensors was determined by maximizing the leak detection for a 2000 samples dataset with different size leaks ($E_c \in [0.1, 2.0]$) randomly generated in the network nodes. Leak detection is implemented by comparing the sample residuals to a threshold, and the sensor location optimum is determined through a genetic algorithm (GA) [37].

*3.2. Realistic Sample Simulation—Uncertainty Modeling*

Taking advantage of the WDN model used for leak location, a group of train and test synthetic datasets were simulated to evaluate the proposed methodology. The hydraulic simulation software EPANET 2.0 [27] is used. In order to make realistic enough simulations of the network modes, the following modeling considerations were made:

- Minimum night flow regime is assumed, ranging from 2 a.m. to 6 a.m. At this time at night, variation in the demand patterns from consumers are very small, therefore, fixed nominal demands can be assumed. The characteristic reduction in demand variation at night time simplifies the location task, increasing its performance; however, this also means that the leak location can only be identified at night.
- Sensor sampling time is considered to be 15 min per sample, with 4 samples in one hour. This amounts to a total of 16 samples in a day under minimum night flow regime. An average between hourly samples is then calculated to filter uncertainties in the data, resulting in 4 filtered samples in a day. A leak scenario is defined as a set of 4 filtered samples from a single day, all of which are generated under the same conditions (leak location and size).
- The leak size for every scenario is sampled randomly from a uniform distribution within the interval $E_C \in [0.5, 1.0]$. This results in leaks that range from 2.6 to 6.3 lps. A timely location and maintenance of leaks of this size can save from 1.1 to 2.6% of the network's total nominal demands during minimum night flow regime. The

leak size range selected could be described as matching medium sized leaks. Leaks higher than that interval are relatively easy to diagnose and sometimes result in pipe bursts visible at street level, which are most likely to be diagnosed by consumer reports. Leaks smaller than the selected interval are, however, mostly background leaks, which are often undetectable due to the uncertainties in network demands and measurements. Figure 3 shows the effect of different sized leaks on pressure head values for every node of the network, which are compared to nominal (no leak) patterns; no uncertainties are considered.

- In real WDNs, actual node demands differ from modeled nominal demands due to variations in consumer patterns. These demand uncertainties must be taken into account in order to achieve realistic simulations. Therefore, node demands were sampled from the following Gaussian distribution: $\aleph \sim \{d_n, \psi d_n\}$, where $d_n$ is the nominal demand and $\psi \in \Re^+$ is a demand uncertainty coefficient. Several uncertainty coefficients are explored in this work. Demand uncertainty is sampled as a Gaussian distribution centered around the nominal demand because the latter has been estimated as the most probable value. Furthermore, by defining demand uncertainty standard deviation as a factor of the nominal demand, nodes with higher nominal demand values are modeled with higher uncertainty.

- In order to guarantee a realistic simulation of the pressure sensors, measurement noise is sampled from the following uniform distribution $[-0.025, 0.025]$ mH$_2$O, and stacked additively with the pressure values simulated.

Figure 4 presents a comparison between pressure head values under nominal (no leak) and leak conditions for every node of the network; consumer demand uncertainty of 10% ($\psi = 0.10$) and measurement noise of 0.025 mH$_2$O are considered for the leak samples, however, no uncertainty is considered for the nominal values. For certain leak nodes, pressure patterns are practically undistinguishable from no leak behaviour, specially for small sized leaks; this fact, combined with the typically limited number of sensors installed in the network, represents the main challenge in the leak location identification task.

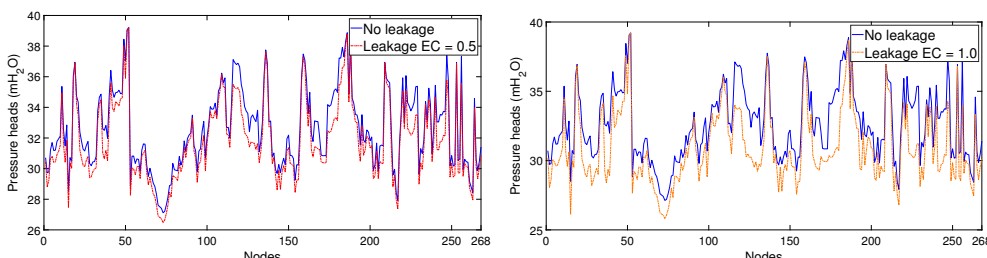

**Figure 3.** Comparing nominal and leak pressure head behaviour.

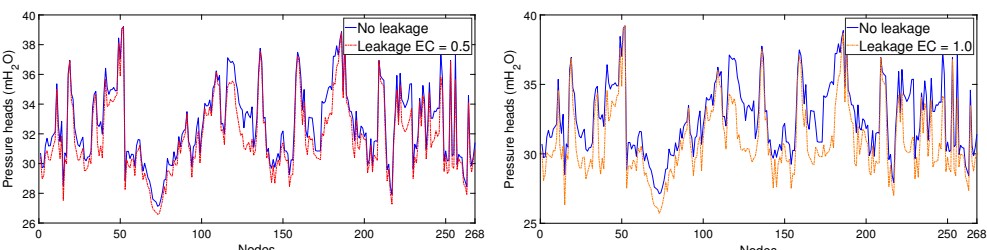

**Figure 4.** Comparing nominal and leak pressure head behaviour with demand uncertainty and measurement noise for leaks.

### 3.3. Experimental Design

#### 3.3.1. Dataset Generation

In order to test the proposed methodology, 5 test datasets were generated following the procedure described above with 5 different uncertainty coefficients: $\psi = \{0.05, 0.075, 0.10, 0.125, 0.15\}$.

Each test dataset contains 2 scenarios per network node, for a total of 536 scenarios (2144 samples) per dataset. A training dataset was also generated to train the SVM multiclass classifier with an uncertainty coefficient of 0.10. It contains 50 leak samples per network node for a total of 13,400 samples.

### 3.3.2. Hybrid Methodology Implementation

The procedure to follow in order to implement the proposed methodology is presented below; this procedure can be applied for any WDN that meets the requirements presented in Section 2.2:

1. The WDN is partitioned into subzones to form classes.
2. A classifier is trained using the training dataset presented in Section 3.3.1 and the labels generated in step 1.
    2.1. For the SVM classifier in particular, hyperparameters $\{\gamma, C\}$ are selected using grid search and 10-fold cross validation in the training set, with the following grid: $\gamma, C \in 2^{\alpha}, \alpha \in \{-2, 5\}$.
3. The number of classes ($n_z$), and zones, in the network is selected to guarantee a 95% classification accuracy for the multiclass classifier.
4. Regarding the TDE algorithm, the mutation ($F$) and crossover ($C_r$) factors are also selected through grid search with the following grid: $F, C_r \in \{0.1 + 0.2\alpha\}, \alpha \in \{1, 2, 3, 4\}$.
5. The number of dominant sensors $n_{sd}$ is selected from the following values: $n_{sd} \in \{2\alpha\}$, $\alpha \in \{1, 2, \ldots, \frac{n_s}{2}\}$, aiming to maximize leak location accuracy; if location accuracy values are similar for different dominant sensor values, the number of dominant sensors that achieves the lowest computational cost will be selected.
6. The stop criteria $g_{max}$, $S_L$, and $\varphi$, as well the population size $K_p$, for the TDE algorithm are selected aiming to minimize the computational cost without resulting in loss of accuracy.
7. Temporal reasoning time horizon $T_H$ is selected according to the sampling frequency and the number of samples in a minimum night flow regime scenario.
8. Neighbor expansion distance $\lambda$ is selected depending on the desired zone size.

Once online, the TDE algorithm is run 5 times for every leak sample, and the candidate solution with the smallest cost function value is selected as the estimated leak node. These multiple runs of the algorithm aim to avoid engaging potential local optima caused by the random nature of the initialization step.

Parameter search for the Modena WDN case study resulted in the following values:

- Classifier hyperparameters: $\{\gamma, C\} = \{4, 8\}$.
- Classifier number of zones: $n_z = 5$.
- Mutation factor: $F = 0.7$, and crossover factor: $C_r = 0.9$.
- Population size: $K_p = 10$.
- Maximum generations: $g_{max} = 500$.
- Stagnation iterations: $S_L = 15$.
- Cost tolerance: $\varphi = 0.05$.
- Pipe distance for neighbors: $\lambda = 250$ m. An average zone size $\overline{ZNS}$ of approximately 6 nodes for the $\psi = 0.05$ case was the desired zone size.
- Temporal reasoning time horizon: $T_H = 4$. All 4 samples in each scenario were selected for temporal reasoning for both the classifier and the TDE steps of the methodology.
- Dominant sensors: $n_{sd} = 4$.

However, any value within the range $\{\gamma, C\} \in [2, 64]$ of the grid can be selected for the Modena WDN, since similar performance values were obtained. Likewise, similar performance values were achieved for $F, C_r \in [0.5, 0.9]$.

### 3.3.3. Comparing Leak Location Strategies

The results achieved by the proposed hybrid method for the 5 test datasets are compared to the results achieved by the TDE and SVM multiclass classifier approaches applied

independently as standalone methodologies. The standalone TDE approach is applied without topological searchspace reduction, considering $\mathbf{Z}_C = [1, N_T]$ as the complete network for every sample. In this case, all the sensors installed are considered as dominant sensors. Hyperparameters and stop criteria for the standalone TDE approach are the same as the ones selected for the hybrid approach. For the standalone multiclass classifier approach, $\mathbf{Z}_C$ is considered as the final estimated leak zone. In this case, $n_z = 35$ is selected such that the average zone size is similar to the one obtained with the hybrid approach. Hyperparameters are identical to the ones selected for the hybrid approach.

## 4. Results and Discussion

A comparison between the hybrid strategy presented and the two independent methodologies is developed according the performance measures defined in Section 2.5. Table 1 and Figure 5 present a location accuracy (*ACC*) comparison between the three methodologies tested under different demand uncertainty percentages ($100\psi$) with the five test datasets.

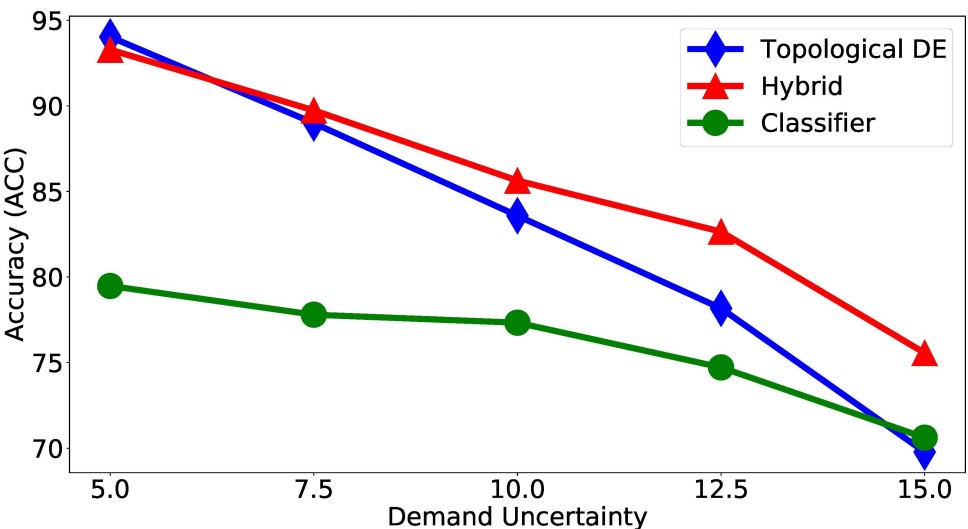

**Figure 5.** Comparing Location Accuracy.

**Table 1.** Comparing Location Accuracy (%).

| Method | Demand Uncertainty | | | | |
|---|---|---|---|---|---|
| | **5%** | **7.5%** | **10%** | **12.5%** | **15%** |
| Classifier | 79.48 | 77.80 | 77.33 | 74.72 | 70.62 |
| TDE | **94.03** | 88.99 | 83.58 | 78.17 | 69.78 |
| Hybrid | 93.28 | **89.74** | **85.63** | **82.65** | **75.56** |

The hybrid approach outperforms the other two methodologies for higher uncertainty values, with a nearly 5% higher location accuracy when the demand uncertainty coefficient is $\psi = 0.15$. However, for the lower uncertainty levels, its performance is similar to the standalone TDE approach, reaching both 93–94% leak location accuracy when the demand uncertainty coefficient is $\psi = 0.05$. The standalone classifier presents the overall lowest performance; however, it deteriorates the least as uncertainty increases, being the most robust to demand uncertainty among the three. It is this robustness in the first stage, combined with the non-dominant sensor measurements deprecation, what makes the hybrid approach more robust, and overall more accurate, than the standalone TDE approach.

All three methodologies are compared regarding the size of the estimated leak location zones, considering size in number of nodes (Figure 6) and total pipe distance (Figure 7). A

smaller estimated leak zone according to both indexes is desirable, and the best zone size results are highlighted in Tables 2 and 3. Both zone size indexes present the standalone classifier generating estimated leak zones nearly two times bigger than the other two methods. Wilcoxon's signed ranked test [38], with a 95% confidence interval, was applied to compare zone size, in number of nodes, between the hybrid and the standalone TDE approaches. This test showed that there were significant statistical differences between both methods for every uncertainty value, with the hybrid approach generating smaller zones. The same statistical test was carried out between the same two methodologies comparing zone size in total pipe distance instead. In this case, the hybrid approach showed significantly smaller zones for all uncertainty values, except for the 0.05 case, for which no statistically significant difference was found. Overall, these results assert the higher performance of the hybrid approach over the standalone TDE and standalone multiclass classifier, achieving higher accuracy values while estimating smaller zones.

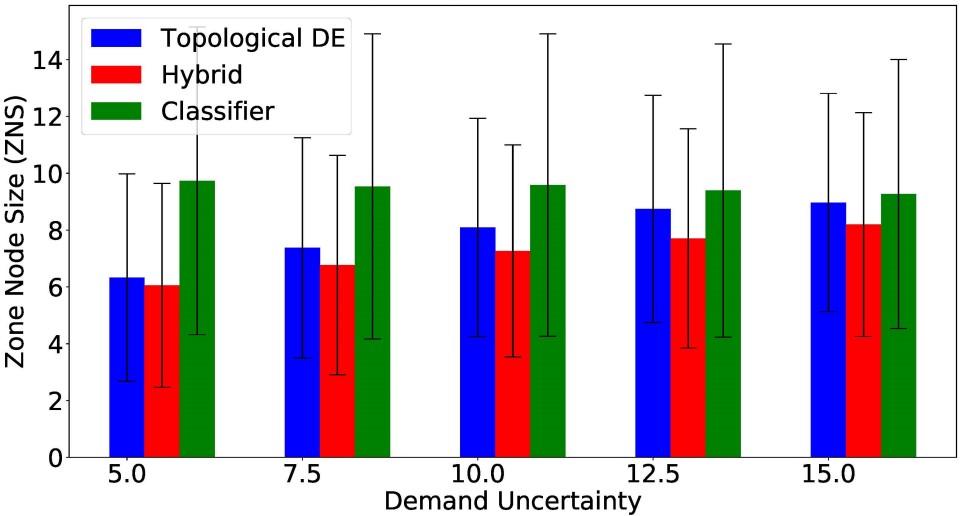

**Figure 6.** Comparing Zone Size in Number of Nodes.

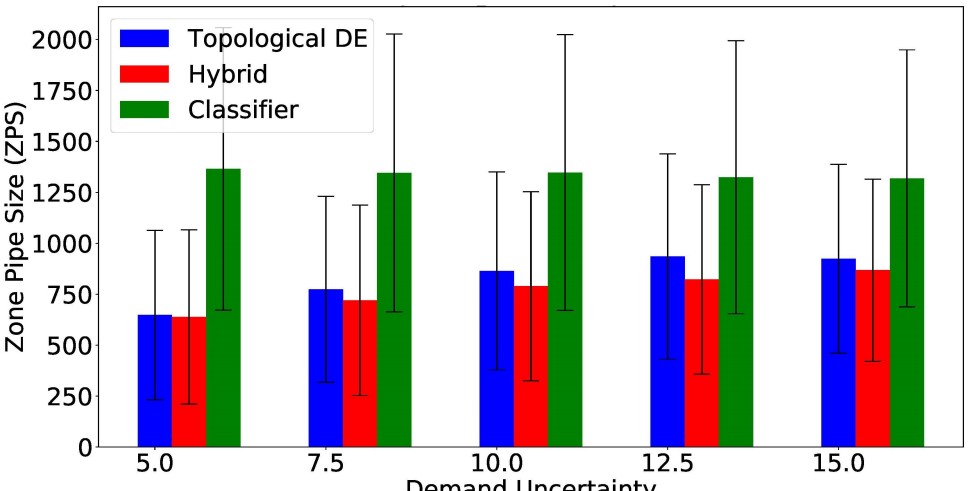

**Figure 7.** Comparing Zone Size in Total Pipe Distance.

**Table 2.** Comparing Zone Size in Number of Nodes $\overline{ZNS}(\sigma_{ZNS})$.

| Method | Demand Uncertainty | | | | |
|---|---|---|---|---|---|
| | **5%** | **7.5%** | **10%** | **12.5%** | **15%** |
| Classifier | 9.73(5.41) | 9.54(5.37) | 9.58(5.32) | 9.39(5.16) | 9.27(4.74) |
| TDE | 6.33(3.65) | 7.38(3.88) | 8.10(3.84) | 8.74(3.99) | 8.97(3.84) |
| Hybrid | **6.06(3.59)** | **6.77(3.86)** | **7.27(3.73)** | **7.71(3.86)** | **8.20(3.94)** |

**Table 3.** Comparing Zone Size in Total Pipe Distance $\overline{ZPS}(\sigma_{ZPS})$.

| Method | Demand Uncertainty | | | | |
|---|---|---|---|---|---|
| | **5%** | **7.5%** | **10%** | **12.5%** | **15%** |
| Classifier | 1365.19(693.03) | 1344.93(681.82) | 1347.05(676.68) | 1324.39(670.07) | 1318.58(630.44) |
| TDE | **647.90(415.12)** | 773.61(456.10) | 864.30(485.54) | 935.00(503.47) | 923.68(463.24) |
| Hybrid | **638.49(427.52)** | **720.27(467.09)** | **789.49(463.95)** | **822.92(464.85)** | **867.75(466.61)** |

Finally, computational cost (*CC*) is compared between the hybrid and the standalone TDE approach in the form of number of generations ($g_{last}$). The computational cost of the classifier approach (both in its standalone version and within the hybrid methodology) is not analyzed since it is comparatively unimpactful. Results are presented in Table 4 and Figure 8.

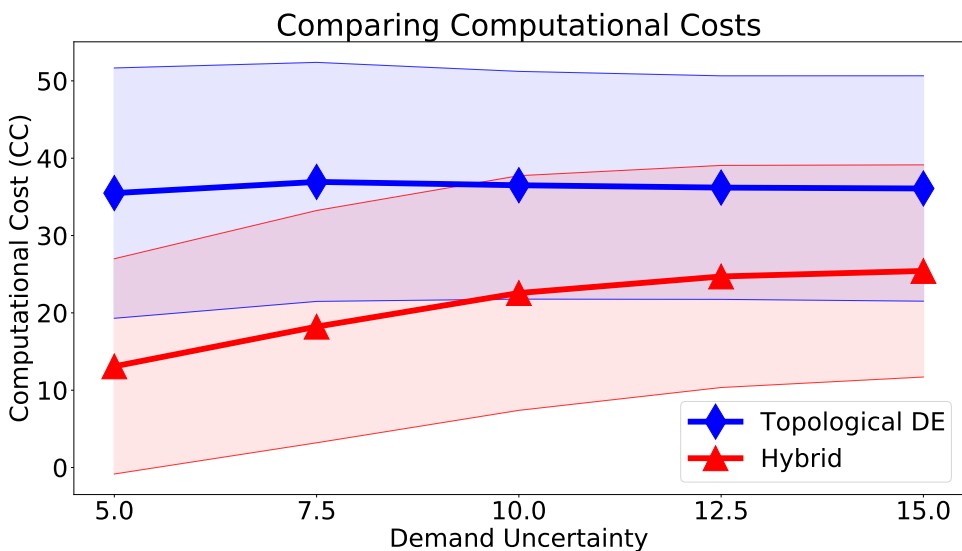

**Figure 8.** Comparing Computational Cost.

**Table 4.** Comparing Computational Cost.

| Method | Demand Uncertainty | | | | |
|---|---|---|---|---|---|
| | **5%** | **7.5%** | **10%** | **12.5%** | **15%** |
| TDE | 35.48(16.18) | 36.94(15.45) | 36.50(14.72) | 36.20(14.45) | 36.08(14.57) |
| Hybrid | **13.08(13.92)** | **18.22(15.02)** | **22.57(15.17)** | **24.71(14.36)** | **25.42(13.71)** |

As expected, by reducing the search space of the inverse problem, the optimization algorithm reaches a solution much faster. The mean hybrid approach TDE run is at least over 30% faster than that of the standalone TDE approach, being as much as 63% faster when the uncertainty coefficient is $\psi = 0.05$. The difference is higher for low uncertainty values and it reaches a plateau as uncertainty increases. The main reason behind this behaviour is the fact that the cost tolerance stop criterion $\varphi$ is selected aiming to guarantee

maximum performance for the lowest uncertainty ($\psi = 0.05$) case. As uncertainty increases, so do variations from the nominal pressure values, increasing cost function values in general. This causes the location of higher uncertainty samples to rarely stop due to the cost tolerance criterion, tilting towards a generation-oriented stop criterion such as stagnation ($S_L$) and max generations ($g_{max}$).

A topological representation of the results obtained for two correctly located leak scenarios is presented in Figure 9. In Figure 9a, the actual leak node is located within the estimated leak zone $\mathbf{Z}_{ngh}$; furthermore, it is located within the leak nodes estimated after applying temporal reasoning to all the samples in the scenario. In Figure 9b however, temporal reasoning would not have sufficed to identify the actual leak node; in this scenario, neighbor expansion guarantees that the actual leak node is inside the estimated zone.

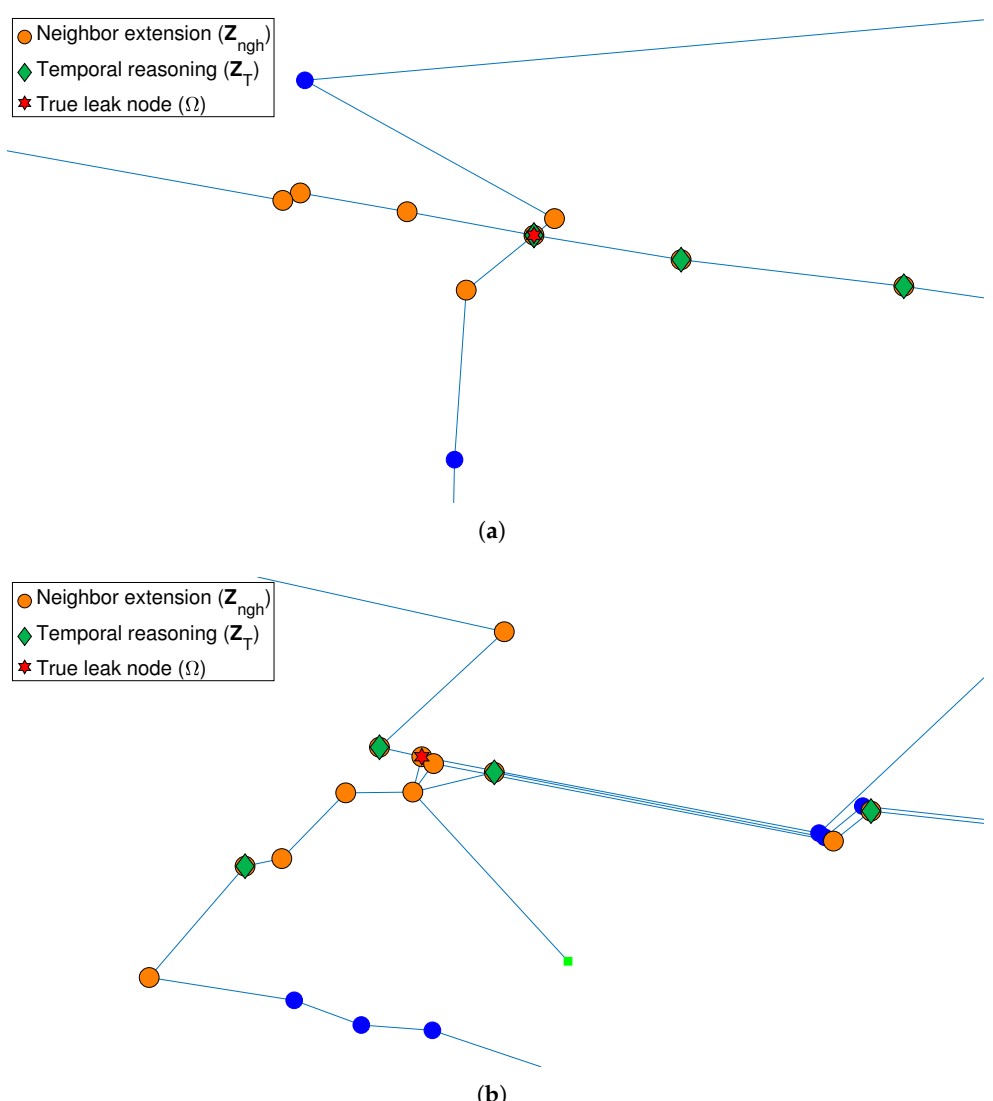

(a)

(b)

**Figure 9.** Accurately located leaks. (**a**) Identified at the temporal reasoning step, (**b**) Identified at the neighbor expansion step.

## 5. Conclusions

This study presents a hybrid methodology for leak zone location in WDNs by combining a multiclass SVM classifier and an inverse problem solved using an optimization algorithm called Topological Differential Evolution (TDE). The classifier is used to reduce the search space of the inverse problem, aiming to increase its performance and reduce its computational cost. From the reduced search space, a set of dominant sensors is selected for leak localization, reducing unwanted uncertainty in measurements. Temporal reasoning is

applied to both the classifier and the TDE, and the estimated leak location is extended to the nearby nodes. In order to evaluate the performance of the proposed strategy, 5 datasets with different demand uncertainty values are tested. The hybrid method is compared to two other methodologies: a standalone SVM classifier and a TDE approach without previous topological search space reduction.

The hybrid approach presented similar location accuracy to the standalone TDE approach for low uncertainty values; however, for higher uncertainty levels, the hybrid approach outperformed the standalone TDE, proving to be more robust with respect to demand uncertainty. The standalone multiclass classifier significantly underperformed in all cases. Estimated zone sizes, measured in number of nodes and total pipe distance, were smaller for the hybrid approach overall in comparison to the other two methods. Regarding computational cost, the search space restriction resulted in a reduction of 30% for high uncertainties and over 50% for low uncertainties. Overall, the hybrid approach demonstrated to be more robust, while reducing the time it takes to locate the leak.

An unintrusive methodology has been proposed with relatively accessible implementation requirements. As long as these requirements have been met, the proposed methodology can be applied to any WDN with a calibrated hydraulic model and it is expected to yield similar results for other medium-to-large WDNs that are meshed in a similar way to the Modena network. Historical data of the network behaviour is not necessary for the implementation of the strategy, however, it can be used if available. Moreover, no specific number of installed sensors is required, and both flow and pressure sensor measurements can be considered simultaneously.

The main limitation to the proposed methodology is the required availability, and proper calibration, of a WDN hydraulic model. Furthermore, the restriction of the leak location task to night hours could result in moderate losses.

Future works will explore different data-driven methodologies in order to improve the performance of the first stage of the hybrid strategy and reduce the topological search space even further; and a study of the performance of the proposed leak location methodology under daytime demand patterns will be performed.

**Author Contributions:** Authors contributions are as follows: Conceptualization, M.J.A.-M., M.Q.-G., C.V. and O.L.-S.; methodology, M.J.A.-M., M.Q.-G. and O.L.-S.; software, M.J.A.-M.; validation, M.J.A.-M.; formal analysis, M.J.A.-M., M.Q.-G. and O.L.-S.; investigation, M.J.A.-M., M.Q.-G. and O.L.-S.; data curation, M.J.A.-M.; writing—original draft preparation, M.J.A.-M.; writing—review and editing, M.J.A.-M., M.Q.-G., C.V. and O.L.-S.; supervision, M.Q.-G., C.V. and O.L.-S.; project administration, O.L.-S. All authors have read and agreed to the published version of the manuscript.

**Funding:** This research was funded by Dirección General de Apoyo al Personal Académico, UNAM.

**Data Availability Statement:** The data sets used for this research are available at https://github.com/mjares/Leak-Data-Simulations-with-Measurement-and-Demand-Uncertainties (accessed on 1 October 2021).

**Acknowledgments:** Marlon J. Ares-Milián and Orestes Llanes-Santiago acknowledge the support provided by National Program of Research and Innovation-ARIA, Project No. 27, CITMA, Cuba.

**Conflicts of Interest:** The authors declare no conflict of interest.

### Appendix A. Dominant Sensor Selection

---

**Algorithm A1** Selecting Dominant Sensor Location

---

{**Inputs**: $\mathbf{S}$: sensors installed in the network, $n_s$: number of sensors installed in the network, $n_{sd}$: number of desired dominant sensors, $\Lambda$: matrix of pipe length distances between nodes, $\mathbf{Z}_C$: zone estimated by the multiclass classifier}

$\mathbf{S}_d = \varnothing$ {*Set of dominant sensors*}

$\bar{n}_{sd} = 0$ {*Number of sensors in the dominant sensor set*}

**while** $\bar{n}_{sd} < n_{sd}$ **do**

  $\mathbf{S}_c = \mathbf{S} \setminus (\mathbf{S} \bigcap \mathbf{S}_d)$ {*Dominant sensor candidates*}

  $\mathbf{S}_{dist} = [\,]$ {*Empty array for sensor distances*}

  **for all** $s \in \mathbf{S}_c$ **do**

    {*For every candidate sensor*}

    $\mathbf{S}_{dist} = \mathbf{S}_{dist} \cup min(\Lambda[\mathbf{Z}_C, s])$ {*Minimum distance between the sensor location s and zone* $\mathbf{Z}_C$}

  **end for**

  **if** any($\mathbf{S}_{dist} == 0$) **then**

    {*If any candidate sensor is inside zone* $\mathbf{Z}_C$}

    $\mathbf{S}_{in} = \mathbf{S}_{c_i}$; *s.t.* $S_{dist_i} == 0$ {*Candidate sensors inside zone* $\mathbf{Z}_C$}

    $\mathbf{S}_d = \mathbf{S}_d \cup \mathbf{S}_{in}$

  **else**

    $S_{close} = \mathbf{S}_{c_i}$; *s.t.* $S_{dist_i} == min(\mathbf{S}_{dist})$ {*Candidate sensor closest to zone* $\mathbf{Z}_C$}

    $\mathbf{S}_d = \mathbf{S}_d \cup S_{close}$

  **end if**

  $\bar{n}_{sd} = |\mathbf{S}_d|$ {*Update number of dominant sensors*}

**end while**

{**Output**: $\mathbf{S}_d$ set of dominant sensor locations}

---

### Appendix B. Topological Differential Evolution Algorithm

A variant of the Differential Evolution (DE) metaheuristic algorithm called Topological Differential Evolution (TDE) [15] is implemented to solve the optimization problem presented in (7). DE is a population-based optimization algorithm that can be used to optimize non-differentiable cost functions. It consists of an initialization step and three operators: mutation, crossover and selection [39]. These operators are applied over a population of solution candidates for a given number of iterations called *generations*. TDE proposes a reformulation of the mutation operator that takes the topological characteristics of the WDN into account.

A population $\mathbf{P}^g = \{\mathbf{x}_1^g, \mathbf{x}_2^g, \ldots, \mathbf{x}_{K_p}^g\}$ is defined for every generation $g$, where each candidate solution $\mathbf{x}_k^g$ is defined as $\mathbf{x}_k^g = \{E_c^g, \omega^g\}$. An overview of the four steps that characterize the Topological DE algorithm are presented below [15]:

1. Initialization

    In the initialization step, an initial population $\mathbf{P}^0$ with $K_p$ candidate solutions is produced by randomly sampling from the search space. The mutation $F \in (0, 2]$ and crossover coefficients $C_r \in (0, 1)$ are also defined at this stage [39]. Initialization is carried out only once for generation 0, while the other three operators are applied to every generation.

2. Mutation

    The mutation operator generates a *mutated* candidate solution $\bar{\mathbf{x}}_k^{g+1} = \{\bar{E}_c^{g+1}, \bar{\omega}^{g+1}\}$ for every member of population $\mathbf{P}^g$ by combining the characteristics of multiple population members. This operator promotes exploration of the search space by generating new characteristics in the mutated population $\bar{\mathbf{P}}^{g+1}$. On one hand, the mutated leak size coefficient $\bar{E}_c^{g+1}$ is generated through classic DE random mutation [39]:

$$\bar{E_c}^{g+1} = E_{c_{r_1}}^g + F(E_{c_{r_2}}^g - E_{c_{r_3}}^g); \tag{A1}$$

where $E_{c_{r_1}}^g$, $E_{c_{r_2}}^g$, and $E_{c_{r_3}}^g$ are the leak size coefficients from randomly selected members in population $g$. These randomly selected population members are different from each other and sampled from a uniform distribution.

On the other hand, the mutated potential leak node, $\bar{\omega}^{g+1}$, is generated following the topology of the WDN [15]:

$$\bar{\omega}^{g+1} = randi(\delta^g), \tag{A2}$$

where *randi* represents the random selection operator which yields a random (sampled from a uniform distribution) network node from the neighbor nodes vector: $\delta^g = \{\delta_1^g, \delta_2^g, \dots \delta_{N_{gh}}^g\}$. A neighbor node $\delta_i^g$ is defined as a node that is connected to $\omega^g$ by a single pipe; and $\delta^g \in \mathbf{Z}_C$ comprehends all the nodes neighboring $\omega^g$ within the topological search space.

3. Crossover

The crossover operator is applied to generation $g$ once the mutation operator has been applied to every member in population $\mathbf{P}^g$. This operator yields a crossed member $\bar{\bar{\mathbf{x}}}^{g+1} = \{\bar{\bar{E}}_c^{g+1}, \bar{\bar{\omega}}^{g+1}\}$ by combining characteristics from the corresponding mutated $\bar{\mathbf{x}}_k^{g+1}$ and pre-mutated $\mathbf{x}_k^g$ population members. This operator is non-linear in nature, and promotes exploitation by combining characteristics from the previous and current generation into the crossed population $\bar{\bar{\mathbf{P}}}^{g+1}$.

Binomial crossover is effected through a crossover probability index $\mathbf{c}_r \in \Re^2$. This crossover probability index is sampled from a uniform distribution for every population member. The crossed candidate is then generated by comparing the probability index to the crossover coefficient $C_r$:

$$\bar{\bar{E}}_c^{g+1} = \begin{cases} \bar{E_c}^{g+1} & if \quad c_{r1} < C_r \\ E_c^g & if \quad c_{r1} \geq C_r \end{cases}, \tag{A3}$$

$$\bar{\bar{\omega}}^{g+1} = \begin{cases} \bar{\omega}^{g+1} & if \quad c_{r2} < C_r \\ \omega^g & if \quad c_{r2} \geq C_r \end{cases}. \tag{A4}$$

4. Selection

The selection operator is applied in order to produce the next generation $\mathbf{P}^{g+1}$ using a greedy criterion [39]. Every crossed member $\bar{\bar{\mathbf{x}}}_k^g$ is compared to its corresponding candidate $\mathbf{x}_k^g$ in population $\mathbf{P}^g$ and the one with the smaller cost function $f_n$ is selected:

$$\mathbf{x}_k^{g+1} = \begin{cases} \bar{\bar{\mathbf{x}}}_k^{g+1} & if \quad f_n(\mathbf{m}_d, M(\bar{\bar{\mathbf{x}}}_k^{g+1})) < f_n(\mathbf{m}_d, M(\mathbf{x}_k^g)) \\ \mathbf{x}_k^g & otherwise \end{cases}. \tag{A5}$$

The three operators are applied to every subsequent generation until any of the following stopping criteria are met:

- Maximum number of generations: $g = g_{max}$.
- Stagnation: the best candidate (the candidate with the smallest value of the cost function $f_n$) $\mathbf{x}_{best}^g$ in the population has remained unchanged for $S_L$ generations.
- Tolerance: the smallest cost function value in the population $f_{n_{best}}$ is smaller than a threshold $\varphi$.

The solution for the optimization problem presented in (7) is then selected as the best member of the last generation $x_{best}^{g_{last}}$. The leak location is then estimated as node $\omega_{best}^{g_{last}}$.

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
