# Peer review of "A Leak Zone Location Approach in Water Distribution Networks Combining Data-Driven and Model-Based Methods"

_water, doi:10.3390/w13202924_

Round 1

Reviewer 1 Report

This study presents interesting approach for the detection of leakage in water supply system.

The characteristics of data-driven methodologies and model-based approaches are well compared. The methods and result are well presented and organized. The authors also considered further applicability of the model in practical use.

There is a minor comment that may improve the manuscript.

In common the leakage in water distribution system is unevenly distributed data. In practice the identification of leakage is also not an easy task. This study used simulated data modeling where the visualization of the input data used for the modeling would be helpful to understand the model performance.

Reviewer 2 Report

The manuscript concerns the important issue of the model-based and data-driven methods, which are commonly used in leak location strategies in water distribution networks. This paper formulates a hybrid methodology in two stages that complements the advantages and disadvantages of data-driven and model-based strategies. The robustness of the method was tested considering measurement and varying demand uncertainty conditions ranging from 5 to 15% of node nominal demands. The performance of the hybrid method was compared to the support vector machine classifier and topological differential evolution approaches as standalone methods of leak location. The hybrid proposal shows higher performance in terms of location accuracy, zone size, and computational load. However, there are a few points that should be addressed to: line 255: Algorithm 1 Selecting Dominant Sensor Location presented in the section 2.4. Stage 2: Leak Location as an Inverse Problem could be presented in the additional appendix. Are there concrete steps that can be recommended and how generalizable are the findings? The region in which the inquiry was conducted, What's distinctive about it? Please indicate what new research brings.The advantages and disadvantages of the proposed method should be indicated. Its limitations and generalized application should be presented. Is this approach was consulted with water and sewer managers?

Round 2

Reviewer 2 Report

Accept in present form.